# Giant *g*-factors and fully spin-polarized states in metamorphic short-period InAsSb/InSb superlattices

Yuxuan Jiang [1,2,5], Maksim Ermolaev[3,5], Gela Kipshidze[3], Seongphill Moon[2,4], Mykhaylo Ozerov [2], Dmitry Smirnov [2], Zhigang Jiang [1] ✉ & Sergey Suchalkin [3] ✉

Realizing a large Landé *g*-factor of electrons in solid-state materials has long been thought of as a rewarding task as it can trigger abundant immediate applications in spintronics and quantum computing. Here, by using metamorphic InAsSb/InSb superlattices (SLs), we demonstrate an unprecedented high value of $g \approx 104$, twice larger than that in bulk InSb, and fully spin-polarized states at low magnetic fields. In addition, we show that the *g*-factor can be tuned on demand from 20 to 110 via varying the SL period. The key ingredients of such a wide tunability are the wavefunction mixing and overlap between the electron and hole states, which have drawn little attention in prior studies. Our work not only establishes metamorphic InAsSb/InSb as a promising and competitive material platform for future quantum devices but also provides a new route toward *g*-factor engineering in semiconductor structures.

Semiconductor structures with large Landé *g*-factors and highly spin-polarized states are pivotal for many quantum device applications. On the one hand, realizing spin-polarized states in high-mobility non-magnetic semiconductors is key to spintronics[1,2]. Spin relaxation in such systems can be largely suppressed, and spin precession can be controlled in a coherent manner. On the other hand, large *g*-factors in strongly spin-orbit coupled (SOC) semiconductors provide the ideal platform for topological-qubit-based quantum computing[3], quantum communication[4], and nonreciprocal spin photonics[5]. Here, the topological phase transition and the interplay of electron spin and photon can be manipulated by the Zeeman energy induced by the external magnetic field.

Conventionally, the *g*-factor engineering in nonmagnetic semiconductors is guided by the Roth formula[6,7]

$$g = g_e - \frac{4\Delta}{3m_0 E_g(E_g + \Delta)}|P_{CV}|^2,  \quad (1)$$

where $m_0$ is the free electron mass, $g_e \approx 2$ is the free electron *g*-factor, $E_g$ is the band gap, $\Delta$ is the SOC energy, and $P_{CV}$ is the momentum matrix element taken between the conduction (CB) and valence band (VB) states. To increase the absolute value of the *g*-factor, high $\Delta$ and $|P_{CV}|^2$, and low $E_g$ are needed, among which the band gap is the easiest to manipulate and can be brought to zero through the use of semiconductor structures such as quantum wells (QWs) or superlattices (SLs). However, this formula fails when $E_g \rightarrow 0$[8], and the relation between the *g*-factor and $E_g$ has not yet been systematically studied in the narrow-gap limit both experimentally and theoretically. When $E_g \rightarrow 0$, the mixing of the CB and VB states inevitably occurs, which suppresses the *g*-factor. Additionally, the electron-hole (e-h) wavefunction overlap is another important factor to consider[9–12]. In QWs and SLs, the electron and hole wavefunctions may center at different locations/layers, resulting in reduced $|P_{CV}|$. As we will show, the strong mixing of the CB and heavy-hole (HH) bands and the reducing e-h wavefunction overlap

[1]School of Physics, Georgia Institute of Technology, Atlanta, GA 30332, USA. [2]National High Magnetic Field Laboratory, Tallahassee, FL 32310, USA. [3]Department of Electrical and Computer Engineering, Stony Brook University, Stony Brook, NY 11794, USA. [4]Department of Physics, Florida State University, Tallahassee, FL 32306, USA. [5]These authors contributed equally: Yuxuan Jiang, Maksim Ermolaev. ✉e-mail: zhigang.jiang@physics.gatech.edu; sergey.suchalkin@stonybrook.edu

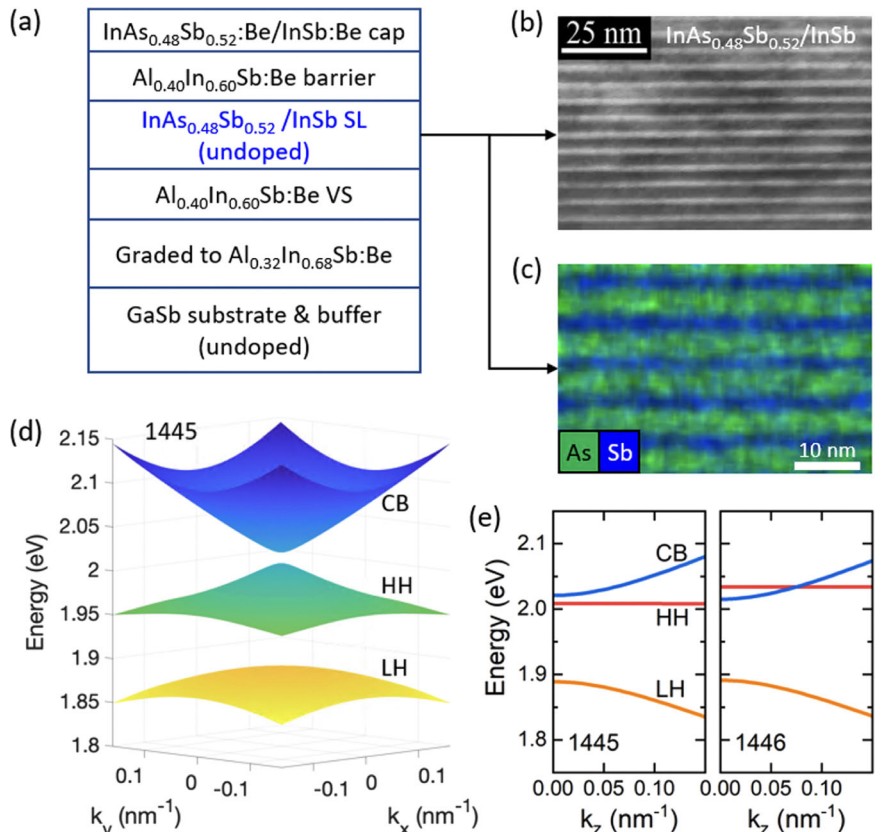

**Fig. 1 | Structural characterization and zero-field band structure calculation. a** Epistructure of sample 1445. **b** Cross-sectional TEM image and **c** EDS image of the SL structure of 1445. **d** The calculated in-plane band structure of 1445 near Γ point.

**e** The calculated energy dispersion along the growth direction ($z$ direction) of both the normal (1445) and inverted (1446) samples.

can explain the limited improvement in the electron $g$-factor in InAs/GaSb QW bilayers despite a small $E_g^{13}$.

To better understand the relation between the band structure (and the wavefunction) and the resulting $g$-factor, we take advantage of the design flexibilities in InAsSb/InSb SLs. The first advantage is the tunable band gaps in InAsSb/InSb type-II SLs via composition ordering on virtual substrate (VS)[14]. It is known that composition ordering, via controlling the atomic order in material's composition profile, can drastically change the electronic and optical properties of semiconductor structures[15–19]. However, spontaneous ordering in InAsSb is difficult to control, and the maximum size of spontaneously ordered domains is reported to be only about 100 nm[20]. Instead, we employ the engineered ordering based on the VS molecular-beam epitaxy (MBE) technique[14], which enables the synthesis of high-quality crystalline materials with a tunable band structure[19,21,22]. The VS technique can also accommodate a large lattice constant mismatch between the semiconductor structure and the (physical) substrate, resulting in intriguing electronic states that otherwise are inconceivable with the conventional pseudomorphic growth.

Second, we can adjust the mixing of the CB and VB states and their wavefunction overlap in InAsSb/InSb SLs by changing the band discontinuity through strain engineering or the SL period[23]. In this work, by changing the period of InAsSb/InSb SLs (that is, the thickness of the InAsSb and InSb layers in one period), we achieve an unprecedented high $g$-factor of $g \approx 104$ within a widely tunable range of 20 to 110 and high spin polarization in a practically accessible magnetic field. Most saliently, by combining the magneto-infrared (magneto-IR) spectroscopy study of such SLs with $k \cdot p$ calculations, we find a four-band model that connects the effective $g$-factor with the material's band parameters at low magnetic fields. Our model analysis shows that the effective $g$-factor and the CB-HH mixing exhibit a concurrent change

with the SL period, resulting from the momentum matrix average between the electron and hole bands. Our work establishes InAsSb/InSb SLs as an ideal platform for $g$-factor engineering and achieving an unprecedented large $g$-factor for future quantum device applications.

## Results

### Material synthesis and characterization
We design and grow a series of metamorphic strain-compensated InAs$_{0.48}$Sb$_{0.52}$/InSb type-II SLs using the VS MBE approach (Methods). Each InAs$_{0.48}$Sb$_{0.52}$ layer is under ~1.1% tensile strain, while each InSb layer is under ~2% compressive strain. The choice of the composition best matches the lattice constant in the VS yet is close to the bowing minimum of the band gap[24]. In the discussion below, we will focus on two samples, 1445 and 1446, with a short period of 4-nm/2.25-nm and 5-nm/2.82-nm, respectively. Additional data on sample 1444 (3-nm/1.69-nm) can be found in the Supplementary Note 1.

Figure 1a shows the epistructure of 1445. The high sample quality is evidenced with the cross-sectional transmission electron microscopy (TEM) image of Fig. 1b and the energy dispersive spectroscopy (EDS) image of Fig. 1c. From the eight-band $k \cdot p$ calculation, we expect 1445 to have a normal band gap while 1446 a slightly inverted one, as shown in the calculated energy dispersion in Fig. 1e. The in-plane band structure near Γ point is also plotted for 1445 in Fig. 1d.

### Experimental results and comparison with $k \cdot p$ model
To accurately determine the band structure, we have performed magneto-IR absorption measurements and compared the experimental results with $k \cdot p$ calculations. Such a combination is an effective approach to study III-V semiconductors, as, for example, shown in ref. 25. Figure 2a, c show the false color plots of the relative transmission $T(B)/T(0)$ of the two SLs as a function of energy and magnetic

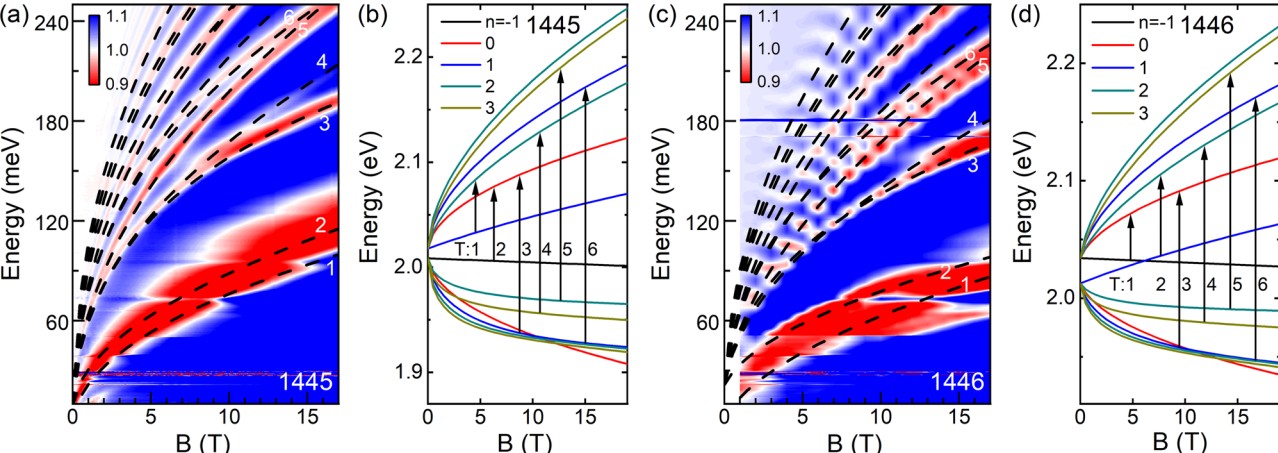

**Fig. 2 | Magneto-IR absorption spectra and the corresponding LL transitions.**
**a** False color map of magneto-absorption at different energies for sample 1445. The black dash lines represent the calculated LL transition energies at different magnetic fields. The first six low-lying transitions are numbered ($T_{1,2,...,6}$) in sequence of their energies. **b** The calculated Landau fan diagram for sample 1445. The LLs are color-coded based on their Pigeon-Brown indices. The black arrows indicate the corresponding LL transitions according to the numbering in **a**. **c**, **d** Same as **a**, **b** but for sample 1446. The seemingly oscillating behavior in **c** is due to the relatively coarse magnetic field grid (1 T step) compared to the 0.12 T step in **a**. All measurements are performed at 5 K.

field. Both samples exhibit a series of spectral dips (i.e., absorptions) blueshifting with increasing magnetic field. These well-developed dips can be attributed to specific inter-Landau-level (inter-LL) transitions in SLs. By tracing the magnetic field dependence of these transitions, we can extract the corresponding LL spectrum and determine the band structure at zero field.

Practically, to extract the material's band parameters from the experimental data, we first calculate the Landau fan diagram of each SL at Γ point using the $k \cdot p$ model. Details of the calculation can be found in the Supplementary Note 2. The magnetic field dispersion of the calculated LLs is plotted in Fig. 2b,d, where the LLs (solid lines) are color-coded based on their Pidgeon-Brown index $n$[26]. The LL transition energy can then be calculated by noting the conventional selection rule $\Delta n = \pm 1$[27], and the first six low-lying transitions are labeled by black arrows in Fig. 2b,d. Next, we can fit all the transitions observed in our experiment with the model calculation. The black dash lines in Fig. 2a,c show the best fits to the data. Excellent agreement with the experiment is achieved. The resulting band parameters (fitting parameters) for the $InAs_{1-x}Sb_x$ alloys with $x = 0.52$ and $x = 1.00$ are listed in the Supplementary Tables 1 and 2. We note that in the fitting process, we employ a single set of band parameters for both SL samples. Hence, their difference in band structure (normal versus inverted) is solely caused by the different layer thicknesses in the SL period.

From the fitting, we also see that the observed absorption dips are mostly due to the interband LL transitions between the CB and HH bands. Both samples have a small band gap, which is 11 meV and 20 meV for 1445 and 1446, respectively. However, the lowest LLs in CB and HH of 1446 cross at around 6 T (Fig. 2d) while the crossing is absent in 1445 (Fig. 2b). Such a crossing behavior is characteristic of an inverted band. Another indication for the band inversion lies in the cyclotron resonance transition $T_1$. The $T_1$ energy in the inverted SL has a weaker magnetic field dependence than that in the normal SL, owing to the stronger mixing of the hole components into the CB. Therefore, we conclude that 1445 is in the normal regime (i.e., positive band gap) while 1446 is in the inverted regime (i.e., negative band gap).

Circular polarization-resolved measurements can further support our $k \cdot p$ model. In the Pidgeon-Brown formalism[26], LL transitions following different selection rules ($\Delta n = \pm 1$) can be distinguished using the right ($\sigma^+$) and left ($\sigma^-$) circularly polarized light[27]. Figure 3 shows the false color plots of the relative transmission $T(B)/T(0)$ of sample 1446 as a function of energy and magnetic field under the excitation of $\sigma^+$ (Fig. 3a) and $\sigma^-$ (Fig. 3b) light. The designed energy range for our broadband

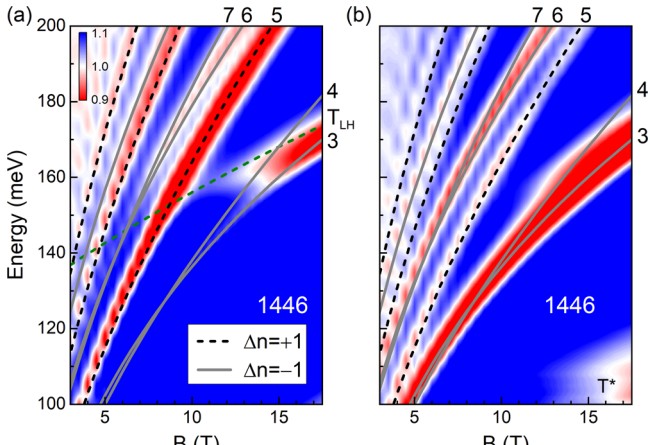

**Fig. 3 | Circular polarization resolved magneto-IR absorption spectra.** False color map of magneto-absorption at different energies for sample 1446 under the excitation of $\sigma^+$ (**a**) and $\sigma^-$ (**b**) circularly polarized light. The dash lines are the calculated LL transitions that follow the selection rule $\Delta n = +1$, while the solid lines represent the $\Delta n = -1$ transitions. The low-lying transitions are numbered in the same sequence as in Fig. 2c, d. $T_{LH}$ denotes a weak transition from LH to HH LLs, and $T^*$ marks a magnetic field-independent spectral feature that we believe is not related to the band structure under this study. All measurements are performed at 5 K.

quarter-wave plate is between 100–155 meV, while considerable polarization contrast is also evidenced between 155–200 meV. Therefore, within this range (100–200 meV), the $\Delta n = +1$ and $\Delta n = -1$ LL transitions are expected to exhibit a stronger absorption in Fig. 3a and b, respectively. Indeed, as one can see from Fig. 3, the $T_3$ and $T_4$ transitions are invisible between 100–155 meV under $\sigma^+$ light but display strong absorption under $\sigma^-$ light. A similar observation can also be made for all the higher LL transitions, particularly $T_5$, $T_6$, and $T_7$. The observed polarization dependence of these transitions agrees well with the theoretical prediction, thus lending strong support to our model.

It is worth noting that the circular polarization-resolved measurements (Fig. 3) may have revealed some spectral features hidden in the unpolarized measurements (Fig. 2). One is the slow changing mode $T_{LH}$ at around 160 meV in Fig. 3a that we attribute to the LL transition from the $n = 0$ LL of the LH bands to the $n = 1$ LL of the HH bands. The other is the (nearly) magnetic field independent dip $T^*$, centered

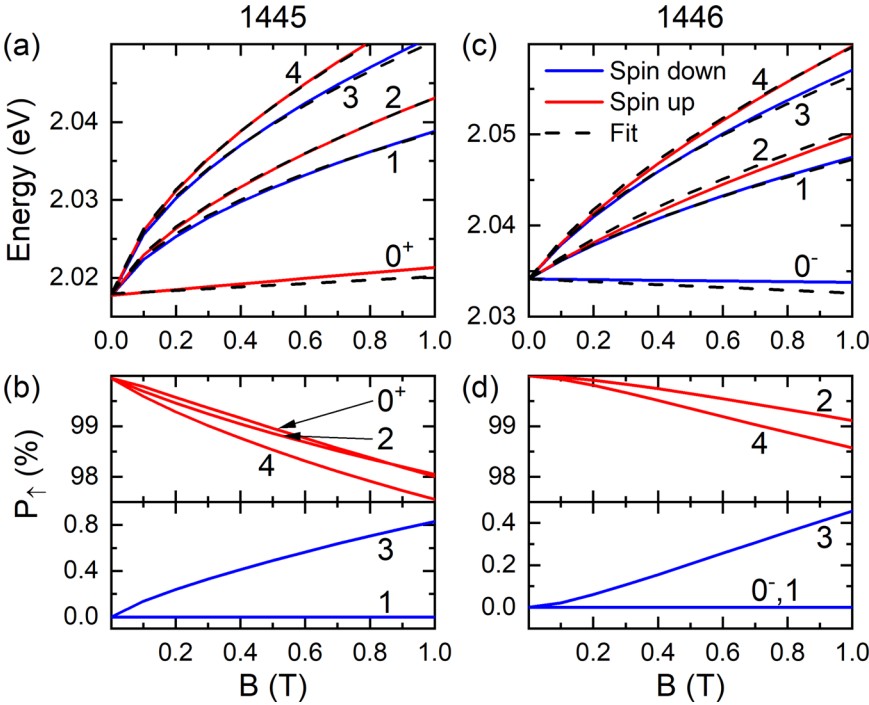

**Fig. 4 | Spin-polarized LLs. a** The calculated low-lying LLs from the *k·p* model (solid lines) for sample 1445 and fit with the four-band model (dash lines) at low magnetic fields. The red and blue colors correspond to the spin-up and down LLs, respectively. **b** The calculated spin up component $P_\uparrow$ in each LL wavefunction. **c, d** Same as **a, b** but for sample 1446. **d** The $0^-$ and 1 LLs are both 100% spin-down polarized and therefore overlap with each other.

around 110 meV in Fig. 3b, which we believe is not related to the band structure under this study. Due to the subtleness of these features, we exclude them from the discussion below.

## Discussion

Next, we discuss the spin polarization and effective *g*-factors in these narrow band gap SLs. We limit the discussion to the CB and its low field behaviors (i.e., $B \leq 1$ T), which is readily accessible in practical applications. The spin polarization of each LL can be obtained by summing up the corresponding spin components in the eigenfunction. The calculation details can be found in Supplementary Note 3. Figure 4a,c duplicate the low-lying LLs of samples 1445 and 1446 at low magnetic fields but color-coded based on their dominant spin polarization. That is, blue and red denote spin down and up, respectively. The corresponding total spin-up component $P_\uparrow$ of each LL is summarized in Fig. 4b,d, where all low-lying LLs exhibit a nearly 100% spin polarization at low fields. We can understand the high spin polarization as the consequence of the decoupling between two spin states[19]. Since the LH and split-off (SO) bands are relatively far away from the CB and do not interact with the HH, the wavefunction of interest is dominated by the CB and HH components. The full *k·p* Hamiltonian around Γ point can then be reduced to two decoupled 2 × 2 diagonal matrices, each of which is spanned by the CB and HH components of the same spin. The low-lying LLs thus preserve a high spin polarization at low magnetic fields.

The high spin polarization in LLs also justifies the discussion of the effective *g*-factor. Conventionally, the *g*-factor of the *m*th LL is defined as[28]

$$g_m^* = \frac{E_m^\uparrow - E_m^\downarrow}{\mu_B B}, \tag{2}$$

where $E_m^{\uparrow\downarrow}$ is the energy of the spin-split LLs and $\mu_B$ is the Bohr magneton. However, the pairing of spin-split LLs may become

ambiguous for narrow-gap materials due to the presence of the unpaired 0th LL. For example, in Fig. 4c, it is not clear whether $LL_2$ should be paired with $LL_1$ or $LL_{0^-}$. A more natural way to consider the band splitting is through a four-band model widely adopted for topological semimetals and insulators[8,29,30]

$$H'(\mathbf{k}) = \begin{pmatrix} M(\mathbf{k}) & Ak_+ & 0 & 0 \\ Ak_- & -M(\mathbf{k}) & 0 & 0 \\ 0 & 0 & M(\mathbf{k}) & -Ak_- \\ 0 & 0 & -Ak_+ & -M(\mathbf{k}) \end{pmatrix}. \tag{3}$$

The basis for InAsSb zinc-blende semiconductor is $[|+,\uparrow\rangle, |-,\uparrow\rangle, |+,\downarrow\rangle, |-,\downarrow\rangle]$[31], where ± denotes the orbitals and ↑↓ the spin directions. Additionally, $\mathbf{k} = (k_x, k_y, k_z)$ is the wave vector, $k_\pm = k_x \pm ik_y$, $A = \hbar v_F$, and $M(\mathbf{k}) = M_0 - M_1(k_x^2 + k_y^2)$. Here, $\hbar$ is the reduced Planck constant, $v_F$ is the Fermi velocity, $M_0$ is related to the band gap $E_g = 2M_0$, and $M_1$ is the parabolic band component arisen from interactions between the CB and other bands[32]. In Supplementary Note 6, we further show that the four-band model can be reduced from the eight-band *k·p* model in the subspace of CB and HH.

In the presence of a magnetic field (along the *z*-direction), the Zeeman effect is included by adding a diagonal matrix $H_Z = \mu_B B[g_0, g_0, -g_0, -g_0]/2$ with $g_0$ being the *g*-factor induced by the remote bands away from CB and VB[8]. The corresponding LLs of the model read

$$\begin{aligned} E_{m=0}^s &= s(M_0 - M_1 k_B^2 + \tfrac{1}{2} g_0 \mu_B B), \\ E_{m\neq0}^s &= s(-M_1 k_B^2 + \tfrac{1}{2} g_0 \mu_B B) + \alpha \sqrt{2\hbar^2 v_F^2 m k_B^2 + (M - M_B)^2}, \end{aligned} \tag{4}$$

where $s = \pm 1$ is the spin index, $\alpha = \pm 1$ is the band index, $k_B = \sqrt{eB/\hbar}$ is the inverse magnetic length with $e$ being the elementary charge, and $M_B = 2m M_1 k_B^2$ is the field-induced gap. Interestingly, besides $g_0$, the $M_1$ parameter leads to a linear-in-*B* LL splitting, which can be translated

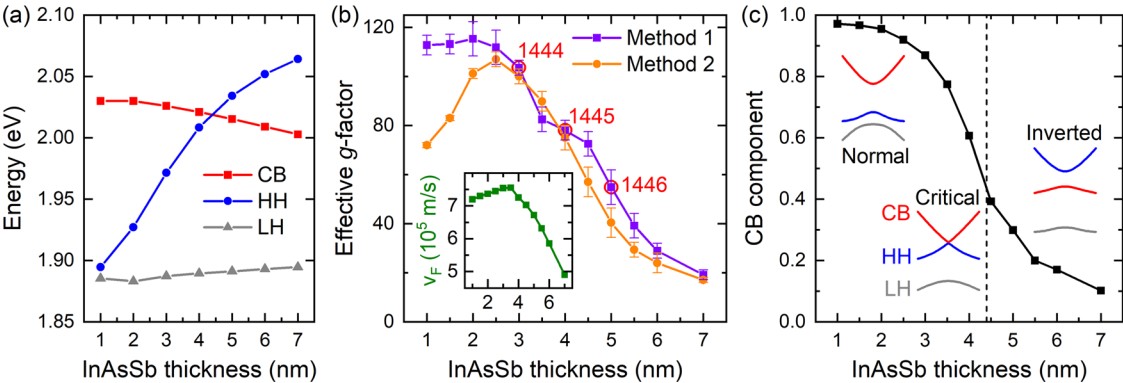

**Fig. 5 | Tunable effective *g*-factors. a** Evolution of the energy bands (CB, HH, and LH) as a function of the SL period ($t_{period} = t_{InAsSb} + t_{InSb} = 1.6t_{InAsSb}$), calculated using the band parameters extracted from the experiment. **b** Evolution of the effective *g*-factors for the CB, extracted using two different methods, as a function of $t_{InAsSb}$. The error bars are determined by varying the Kane energy $E_p$, as explained in the Supplementary Note 5. Red circles label out the three SL samples measured in this work, where $t_{InAsSb}$ = 3 nm, 4 nm, and 5 nm, respectively. Inset: The extracted Fermi velocity $v_F$ from the four-band model as a function of $t_{InAsSb}$. **c** The CB component in the wavefunction as a function of $t_{InAsSb}$. The calculation is performed at 1 T. Insets: Band evolution from the normal state to the inverted state as increasing the SL period. The critical state is indicated by a vertical dash line.

into a *g*-factor by

$$g_1 = -\frac{2M_1 k_B^2}{\mu_B B} = -2\frac{eM_1}{\hbar\mu_B}. \qquad (5)$$

Therefore, we arrive at a total effective *g*-factor $g_{eff} = g_0 + g_1$. Next, we will simply fit the *k·p* results with the four-band model to extract the corresponding $g_{eff}$ values.

Figure 4a, c show the fitting results between the effective Hamiltonian $H'$ of the four-band model (dash lines) and the *k·p* calculations (solid lines) for the low-lying CB LLs of samples 1445 and 1446 at low magnetic fields. Additional fitting can be found in the Supplementary Note 4. As one can see, all the LL splittings can be well described by a single, universal *g*-factor, in contrast to the LL index and *B*-dependent *g*-factors in previous studies of the narrow-gap semiconductors with weak non-parabolicity[28]. Such a *g*-factor is convenient for comparison between different materials.

From the fitting, we can further extract the band parameters in InAsSb/InSb SLs with a fixed thickness ratio (i.e., $t_{InAsSb}/t_{InSb} \approx$ 1.8, which is the case for all three samples studied). In this way, we keep the SL strain compensated while tuning the band structure and e-h wavefunction overlap via changing the period ($t_{period} = t_{InAsSb} + t_{InSb} = 1.6t_{InAsSb}$). The extracted band parameters can well describe the electronic structure of all three samples measured and predict the properties of similar SLs with different periods. The resulting band edge, effective *g*-factor, and Fermi velocity are shown in Fig. 5a, b as a function of $t_{InAsSb}$. Specifically, the extracted *g*-factor for samples 1445 and 1446 is 78 and 55, respectively. The error analysis of the *g*-factors is described in the Supplementary Note 5. We note that as the SL period increases, the system changes from the normal to the inverted regime, with a zero band gap occurring at $t_{InAsSb} \approx$ 4.4 nm. For $g_{eff}$ (method 1, defined by Eq. (2) above), it saturates at around 110 in the short-period limit and gradually decreases to 20 across the normal to inverted band transition. The maximum *g*-factor is about twice larger than that in the composite materials, $g_{InAsSb} \approx g_{InSb} \approx 55$[33,34]. We further note that although a large *g*-factor of $g \approx$ 117 was recently proposed in $InAs_{0.4}Sb_{0.6}$[35] using the Roth formula with very optimistic band parameters, our work presents the first experimental realization of $g \approx$ 104 (in sample 1444, Supplementary Note 1) in nonmagnetic semiconductors.

Moreover, the ability to tune *g*-factors in a wide range is intriguing. We can attribute such a tunability to the wavefunction mixing between the CB and HH bands. To understand this, one can consider a four-band model with CB and HH components only. Their LLs are then strictly spin degenerate. To introduce spin splitting, the wavefunction must mix with additional band characters such as the LH and SO bands. In III-V semiconductors, since HH does not interact with LH and SO, one can easily imagine that if the wavefunction has more of the CB component (than the HH component), then it will mix in more of the LH component in a magnetic field, leading to stronger splitting, and vice versa. Using the Kane energy and the band gap between CB and HH, we can estimate the CB component in the wavefunction at 1 T by using a two-band *k · p* model and the result is shown in Fig. 5c. Here, the wavefunction is completely dominated by the CB component in the short-period limit, concurrent with the saturated *g*-factors in Fig. 5b (method 1). As the SL period increases, the HH band moves closer to the CB (Fig. 5a), and eventually, the HH becomes dominant in the inverted regime. Due to the reduced CB-LH interactions in this regime, the *g*-factor is suppressed and can become even lower than that in composite materials. Overall, as shown in Fig. 5b,c, the *g*-factor (method 1) and the CB component follow a similar trend as increasing the SL period and with a comparable degree of reduction. This picture is further supported by the Supplementary Notes 4 and 6, whereby extending the fitting in Fig. 4a,c to a higher magnetic field, we show that $g_{eff}$ is dominated by the $g_1$ (or $M_1$) parameter, and $M_1$ is related to the CB-LH and CB-SO interactions.

Before closing, two additional points need to be made. First, the e-h wavefunction overlap also plays a role in *g*-factor engineering in InAsSb/InSb SLs. It is reflected in the weight of the off-diagonal matrix elements describing the interactions between different bands and hence leads to the modification of the Fermi velocity $v_F$[19,36] and *g*-factor. In the inset to Fig. 5b, we plot the $v_F$ as a function of the SL period and find that the $v_F$ decreases gradually across the normal to inverted band transition. This behavior is consistent with the reduced wavefunction overlap between the CB and HH bands in our SLs across the transition. However, such an effect seems relatively weak compared to the wavefunction mixing effect described above, as the change in the $v_F$ (~35%) is much less than that in the *g*-factor (~83%), and the change in wavefunction overlap between CB and HH is the largest among all across the transition.

Second, the *g*-factor defined above with the four-band model (method 1) is proportional to the $M_1$ parameter, as shown in Eq. (5). Such a *g*-factor is related to the splitting of LLs with the same index (Eq. (2)). However, as the *g*-factor increases, if the Zeeman splitting becomes larger than the energy spacing between adjacent LLs, it leads to an alternative definition (method 2). That is, one can follow a specific LL and define the *g*-factor from the splitting with the closest LL of

## Table 1 | Sample structure information

| Sample | $t_{InAsSb}$ (nm) | $t_{InSb}$ (nm) | $t_{InAsSb}/t_{InSb}$ | $N$ | $L_t$ (µm) |
|---|---|---|---|---|---|
| 1444 | 3 | 1.69 | 1.8 | 213 | 1.0 |
| 1445 | 4 | 2.25 | 1.8 | 160 | 1.0 |
| 1446 | 5 | 2.82 | 1.8 | 147 | 1.15 |

$t_{InAsSb}$ and $t_{InSb}$ are the thicknesses of the InAsSb and InSb layers in each SL period, $N$ is the number of periods repeated in the SL, and $L_t$ is the total thickness of the SL structure. In all samples, InAsSb represents $InAs_{0.48}Sb_{0.52}$.

the opposite spin regardless of its LL index. The extracted *g*-factors using these two methods are summarized in Fig. 5b as a function of the SL period. As one can see, the two methods agree well before entering the short-period limit. The difference in the short-period limit is simply caused by choosing a different LL splitting to define the *g*-factor. Nevertheless, no matter which *g*-factor definition one uses, a giant *g*-factor of $g_{eff} \approx 110$ is expected in our short-period InAsSb/InSb SL samples.

In conclusion, using metamorphic strain-compensated $InAs_{0.48}Sb_{0.52}$/InSb SLs as an example, we demonstrate the feasibility of realizing tunable *g*-factors and fully spin-polarized states in InAsSb ordered alloys. The VS approach, which was used to grow the SLs, allows for large variations of the SL period while keeping the band gap narrow, enabling key ingredients for obtaining a high *g*-factor: strong mixing of the CB and VB states and high e-h wavefunction overlap. We show that in narrow-gap InAsSb/InSb SLs, the effective *g*-factor can be tuned from 20 to 110 simply by changing the SL period. To extract the *g*-factor, we employ a four-band model that is widely used in topological materials to analyze our magneto-IR spectroscopy results and identify the interactions between the CB and LH/SO bands as the dominant source of the LL (spin) splitting. Our work sheds light on how to band-engineer semiconductor structures with large *g*-factors for future quantum device applications.

## Methods

### Sample growth

The $InAs_{0.48}Sb_{0.52}$/InSb type-II SLs were grown by the solid-source MBE on undoped (100) GaSb substrates. The metamorphic buffer was graded from GaSb to $Al_{0.32}In_{0.68}Sb$ with a thickness of ≈2240 nm. The graded buffer was p-doped (Be) to ~$10^{16}$ cm$^{-3}$ and followed by $Al_{0.40}In_{0.60}Sb$ VS with a thickness of 500 nm and an effective lattice constant of 6.33 Å. The strain-compensated $InAs_{0.48}Sb_{0.52}$/InSb SL was grown on top of the VS[14,19].

We focus on two SL samples (1445 and 1446) with different periods in the main text. One has 160 periods; each period includes 4 nm of $InAs_{0.48}Sb_{0.52}$ and 2.25 nm of InSb. The other has 147 periods; 5 nm of $InAs_{0.48}Sb_{0.52}$ and 2.82 nm of InSb in each period. The structure information of these samples, together with that of the third sample (1444) described in Supplementary Notes 1, are summarized in Table 1. We note that the number of periods $N$ is sufficiently large in our SL samples such that the electronic structure and *g*-factor are independent of $N$ but can be tuned by changing the thickness of each SL period. In all samples, the SL is followed by a 200 nm thick $Al_{0.40}In_{0.60}Sb$ top barrier and a cap layer consisting of 4–5 periods of the SL to avoid the oxidation of Al-containing barrier material. The cap and barriers are p-doped (Be) to ~$10^{16}$ cm$^{-3}$ for both samples to compensate for the barrier background doping and avoid the formation of two-dimensional carrier "pockets" at the SL interface with the barrier[37,38]. This compensation doping approach has been implemented in our previous work[24], where no magneto-absorption features from barriers are observed.

Structure studies of similar SL samples can be found in Ref. 21, where it is shown that the SLs exhibit a dislocation-free periodic structure but with some interface roughness and compositional disorder. The effects of interface roughness can be modeled as a local random change in the SL layer thickness. Such a lateral thickness fluctuation does not directly contribute to the LL transition energies but leads to the broadening of the transitions[21]. The broadening is somewhat suppressed in the ternary-binary SLs (InAsSb/InSb) compared with the ternary-ternary ones. In addition, the composition of $InAs_{0.48}Sb_{0.52}$ layer in our SLs is chosen to be close to the minimum of the band gap bowing curve. It makes the material parameters less sensitive to the fluctuations of the composition.

### Magneto-infrared experiment

In this work, the SL samples are measured at 5 K using a Fourier transform IR spectrometer connected to a 17.5 T superconducting magnet through a vacuumed light pipe. The magnetic field is applied along the growth direction (*z* direction) of the SLs, and the transmitted light is collected by a Si bolometer mounted beneath the sample. By adding a linear polarizer and a broadband quarter-wave plate in the light path, we can perform circular polarization-resolved measurements, which further help to distinguish the optical transitions between different selection rules[39,40].

## Data availability

The data that support the findings of this study are available from the corresponding author upon reasonable request.

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

## Acknowledgements

This work was primarily supported by the NSF (Grant nos. DMR-1809120 and DMR-1809708). The MBE growth at Stony Brook was also supported by the U.S. Army Research Office (Grant no. W911NF2010109) and the Center of Semiconductor Materials and Device Modeling. The magneto-IR measurements were performed at the National High Magnetic Field Laboratory (NHMFL), which is supported by the NSF Cooperative agreement no. DMR-1644779 and the State of Florida. Y.J. acknowledges support of the NHMFL Jack E. Crow Postdoctoral Fellowship. Y.J., D.S., and Z.J. acknowledge support from the DOE (for magneto-IR) under grant no. DE-FG02-07ER46451. We thank A.A., G.B., and B.L. for helpful discussions.

## Author contributions

S.S. and Z.J. conceived the idea. G.K. performed the sample growth. M.E. performed the EDS and TEM measurements. M.E., S.M., D.S., and M.O. performed magneto-IR experiment. Y.J. performed the $k \cdot p$ model calculation and analyze the data. Y.J., Z.J., and S.S. wrote the paper. All authors contributed to the discussion and revision of the paper.

## Competing interests

The authors declare no competing interests.
