## [Peer Review File · Nature Communications]

Reviewers' Comments:

Reviewer #1:

Remarks to the Author:

Yuxuan Jiang et al. report the study of metamorphic short-period InAsSb/InSb superlattices structures by using magneto-IR spectra and model with the KP model. In such superlattices quantum wells, the authors found a tunable and exceptional high g-factor up to 110. To extract the effective g-factor, the authors performed the magneto-IR absorption measurements of three MBE grown InAsSb/InSb superlattices with different thickness ratios of InAsSb and InSb. By using the eight-band KP model, the authors fit the magneto-IR spectrum as the result of the inter Landau Level (LL) transitions. To justify the KP model, the circular polarized magneto-IR spectra are conducted to check the expected selection rules of the LL transitions. Two reported samples in the main text show different band structures which can be justified by the KP model and absorption spectrums. To deduce the effective g-factor and other band parameters, the authors introduce a four-band model allowing them to fit the low magnetic field regime of the KP model. From the fitting, the energy gap, effective g-factor, and Fermi-velocity can be extracted for different InAsSb thicknesses with a fixed thickness ratio of 1.8 (InAsSb to InSb). Benefitting from the four-band model, the authors avoid the issue of the Roth formula which fails at energy gap equals zero. From the result, the energy gap changes monotonically from 120 meV to -40 meV. Furthermore, the g-factor and Fermi velocity have similar behavior showing higher values for smaller InAsSb thickness regions then decreasing to a minimum after the inversion of the band.

InAsSb is an interesting ternary material with its potential for photodetector, spintronic device, and topological system due to large spin-orbit coupling. In the direction of the photodetector, InAs/InAsSb and InSb/InAsSb superlattice structure were both studied also by the co-authors (ref. 21). The band inversion due to superlattice structure was reported by coauthors in ref. 22. For the ternary InAsSb material, Ref. 33, Joachim E. Sestoft. et al. (Phys. Rev. Mater. 2, 044202, 2018), and Mayer W. et al. (ACS Appl. Electron. Mater. 2020, 2, 2351–2356) have studied several different compositions of InAsSb. In other words, both the InSb/InAsSb superlattice and InAsSb ternary may not be a novel subject. However, the authors may consider such an InSb/InAsSb superlattice structure as the applications in the electrical transport to maximize the uniqueness of the large-g factor and strong spin-orbit coupling. To this end, there will remain issues due to the superlattice structure such as non-localization of the wave function, g-factor anisotropic, and weakening of spin-orbit coupling with superlattice. Besides, I still have several concerns about the manuscript.

Main questions:

1. The large g-factor of ternary material has been discussed in the reference, Mayer W. et al. (ACS Appl. Electron. Mater. 2020, 2, 2351–2356). In table I, it has been mentioned that by barely considering the Roth formula, the g-factor of InAsSb can be as high as -116 (in a quantum well case, there is no issue using the Roth formula). Although it is a rough estimate, by comparing this value to the reported result, the reported effective g-factor of 110 is not surprisingly high.
2. One of the novelties may remain from tuning the g-factor based on InAsSb thickness with a fixed ratio to InSb. In the earlier report, the tuning is mainly based on the concentration of As and Sb. However, one thing that remains puzzling to me is that the lowest value of the g-factor is around 30, much lower than both InSb and InSbAs. This is indeed a very interesting point that may suggest that the e-h wave function may play an important role which is rarely considered in the past. It may be worth elaborating and understanding the physics behind this low value.
3. The validation of the KP model? It is strange to me that the authors use a single set of fitting parameters for all the samples. The normal (1445) and inverted (1446) bandgap are different as the authors pointed out. Using the same band parameter for all three samples is still strange to me. Indeed this simplifies the story to only the variation of InAsSb thickness as discussed in SI note 2. Since the entire story is built on the KP model, it may be worth exploring more parameter space to make sure it is valid.
4. Furthermore, in the SI table, I list the band parameter. It is stated in the SI note 2 " we start with the parameters using the interpolation scheme between InSb and InAs but adjust the

parameters in Tab. 1 to best fit the experiment. " However, there are many parameters which can be adjustable. As mentioned, the entire story is based on the KP model, if there are multiple solutions to this fits, it may change the story. Especially, the delta (spin-orbit coupling) energy for InAsSb is expected to be higher than InSb and InAs (Mayer W. et al. (ACS Appl. Electron. Mater. 2020, 2, 2351–2356), Ref. 18 and Ref. 33). It is contradicting table I. Moreover, ref. 6 in the SI has provided a table for InAsSb band parameters which is very different from the values of the fitting. Could the authors elaborate more on this point?

5. The effective g-factor is obtained from fitting to a model instead of experimental data. It was surprising to me in the first place. But, I later realized that fitting the four-band model to the magneto-IR spectra will lead to very large errors due to not clearly defined transition between LLs. Is this due to the temperature effect? Did the authors also measure the magneto-IR spectra (figure 3) at 5K? What about the error bars to this fitting? Would it be possible to estimate the errors from the experiment data set when fitting with 8-band KP model?

6. The result suggests that M1 is the dominant factor for the large effective g-factor. This is from the overlapping of CB and HH which makes sense to me around the small gap and HH is close to CB. Based on the assumptions, the model should be limited to a small gap with a small magnetic field which suggests around the normal-inverted band transition. However, the large effective g-factor is also observed for a normal band with a very large bandgap up to 120 meV. The large g value remains the same for the Eg from 120 to 30 meV. This is beyond the assumptions of the four-band mode.

7. Lastly, it is also possible to directly measure the effective g-factor from the electrical transport properties. For example, Qu, F., et al. Nano Letters, 16(12), 7509–7513, Zijin Lei, et al., Phys. Rev. Research 3 023042 (2021) and ref. 33 etc. can measure the effective g-factor from the energy difference of the spin split LLs. It will provide a piece of more direct evidence to the claimed large g-factor. On the other hand, several potential applications of such superlattice structure as the authors mentioned in the main text will require electrical transport. Therefore, an electrical transport result may prove the usefulness of the InSb/InAsSb superlattice.

Other suggestions:

1. In Fig. 3b, sample 1446 shows an oscillating behavior in the energy direction between 60-240 meV. This is not seen in other samples. Any reasons for that?
2. SI table I needs units.
3. The labels in figure 4 are not very clear. I would replace green labels with another color.

Reviewer #2:

Remarks to the Author:

This paper by Y. Jiang et al. reports the realization of giant Lande g-factor (104-110) in InAsSb/InSb SLs, which are designed/explained based on the virtual substrate technique, band gap tuning, and electron-hole wavefunction overlaps. The large g-factor is generally of great interest in a variety of applications including quantum devices as described in the abstract. The sample system and experimental/theoretical schemes seem to have solid basis, which are extensions of previous works done by some of the authors. I found, however, several problems in the manuscript; one of which would be critical and make the judge of significance difficult. Some of the points are listed below. Therefore, I do not recommend this paper to be published in Nature Communications in the present form.

1. It is critical to add the error bars in the evaluation of g-factor, especially when the authors are claiming the record-large value. The error can appear from experimental broadening of magneto-IR absorption (apparent in presented figures), fitting procedure, and also through theoretical models. Please define the error and discuss the possible range of g-factor value in the manuscript.

2. Please revise the right panel of Fig. 1, and depict quantitatively how much wave-function overlap is expected at which SL period.

3. I believe that "high-resolution cross-sectional TEM" nowadays means that with atomic resolution. Please replace Fig. 2 with the one showing the atomically sharp interface. If some disorder is unavoidable in the growth, add some discussion on the possible effects of disorder in the g-factor engineering performed in this work.

4. Please explain Fig. 5(d) (actually not labeled) in more detail. Why the g-factor does not change at all for the InAsSb thickness of 2 to 3 nm (also for 5.5 to 6 nm) even when the bandgap keeps changing?

RESPONSE TO THE REVIEWERS

Reviewer #1 (Remarks to the Author):

Yuxuan Jiang et al. report the study of metamorphic short-period InAsSb/InSb superlattices structures by using magneto-IR spectra and model with the KP model. In such superlattices quantum wells, the authors found a tunable and exceptional high g-factor up to 110. To extract the effective g-factor, the authors performed the magneto-IR absorption measurements of three MBE grown InAsSb/InSb superlattices with different thickness ratios of InAsSb and InSb. By using the eight-band KP model, the authors fit the magneto-IR spectrum as the result of the inter Landau Level (LL) transitions. To justify the KP model, the circular polarized magneto-IR spectra are conducted to check the expected selection rules of the LL transitions. Two reported samples in the main text show different band structures which can be justified by the KP model and absorption spectrums. To deduce the effective g-factor and other band parameters, the authors introduce a four-band model allowing them to fit the low magnetic field regime of the KP model. From the fitting, the energy gap, effective g-factor, and Fermi-velocity can be extracted for different InAsSb thicknesses with a fixed thickness ratio of 1.8 (InAsSb to InSb). Benefitting from the four-band model, the authors avoid the issue of the Roth formula which fails at energy gap equals zero. From the result, the energy gap changes monotonically from 120 meV to -40 meV. Furthermore, the g-factor and Fermi velocity have similar behavior showing higher values for smaller InAsSb thickness regions then decreasing to a minimum after the inversion of the band.

InAsSb is an interesting ternary material with its potential for photodetector, spintronic device, and topological system due to large spin-orbit coupling. In the direction of the photodetector, InAs/InAsSb and InSb/InAsSb superlattice structure were both studied also by the co-authors (ref. 21). The band inversion due to superlattice structure was reported by coauthors in ref. 22. For the ternary InAsSb material, Ref. 33, Joachim E. Sestoft. et al. (Phys. Rev. Mater. 2, 044202, 2018), and Mayer W. et al. (ACS Appl. Electron. Mater. 2020, 2, 2351–2356) have studied several different compositions of InAsSb. In other words, both the InSb/InAsSb superlattice and InAsSb ternary may not be a novel subject. However, the authors may consider such an InSb/InAsSb superlattice structure as the applications in the electrical transport to maximize the uniqueness of the large-g factor and strong spin-orbit coupling. To this end, there will remain issues due to the superlattice structure such as non-localization of the wave function, g-factor anisotropic, and weakening of spin-orbit coupling with superlattice. Besides, I still have several concerns about the manuscript.

We thank Reviewer #1 for the detailed and precise summary of our work, and we appreciate the overall constructive comments.

Main questions:

1. The large g-factor of ternary material has been discussed in the reference, Mayer W. et al. (ACS Appl. Electron. Mater. 2020, 2, 2351–2356). In table I, it has been mentioned that by barely considering the Roth formula, the g-factor of InAsSb can be as high as -116 (in a quantum well case, there is no issue using the Roth formula). Although it is a rough estimate, by comparing this value to the reported result, the reported effective g-factor of 110 is not surprisingly high.

[Our reply]: Indeed, based on the Roth formula and the parameters in Ref. [24] of the main text, we could obtain a g-factor as high as 116. However, as Reviewer #1 points out, such a g-factor is only a rough estimation. First, the estimated g-factor depends on the value of the effective mass (m^*) in use, which usually exhibits a large uncertainty. Taking the recommended value $m^* = 0.0103 m_0$ (where m_0 is the free electron mass) for InAs_{0.4}Sb_{0.6} from Ref. [S6] of the Supplementary Information (SI), we get $g \approx 66$. Second, another source of uncertainty is the bowing effects in ternary alloys. It was experimentally demonstrated in [Ref-1] that the Kane energy E_p has a nonzero bowing in InAsSb and reaches the value around 15-16 meV for the Sb composition near 0.6. (We note that the bowing effect in E_p has largely been neglected in previous studies.) Using this E_p value, we arrive at $g \approx 80$ in InAs_{0.4}Sb_{0.6}. The combination of these two issues makes the g-factor even smaller. Therefore, we argue that $g \approx 116$ is a very optimistic value when the bowing effect in E_p is neglected.

On the other hand, our k.p-based method is built upon a consistent set of band parameters (to be explained in reply to Comment (4) below) and takes account of the bowing effects. In this work, we directly fit the experimental data to determine the band parameters, and the deduced g-factors are thus more accurate. The error bars of the deduced g-factors will be discussed below. To the best of our knowledge, this is the first *experimental* work that demonstrates the effective g-factors in InAsSb superlattices (SLs) larger than the values in bulk alloys.

[Ref-1] S. N. Smith *et al.* Semicond. Sci. Technol. **7**, 900 (1992).

[Our changes]: We have included a discussion about the Mayer *et al.* 2020 paper in the revised manuscript (page 11)

“”

2. One of the novelties may remain from tuning the g-factor based on InAsSb thickness with a fixed ratio to InSb. In the earlier report, the tuning is mainly based on the concentration of As and Sb. However, one thing that remains puzzling to me is that the lowest value of the g-factor is around 30, much lower than both InSb and InSbAs. This is indeed a very interesting point that may suggest that the e-h wave function may play an important role which is rarely considered in the past. It may be worth elaborating and understanding the physics behind this low value.

[Our reply]: We thank Reviewer #1 for raising this point. The relatively low g-factor in the wide-period limit (inverted band structure) is indeed interesting and worth further

elaboration. We can understand it from wavefunction mixing.

If we consider a k.p model with only four bands, that is, spin-up and spin-down conduction (CB) and heavy-hole (HH) bands, and their interactions, then the Landau levels (LLs) are always degenerated for index $n > 0$ and do not exhibit any splitting. To introduce splittings between LLs, one needs to include the interaction with more bands, such as the light-hole (LH) and split-off (SO) bands. For simplicity, here we consider the LH band. Therefore, we can qualitatively relate the LL splittings with the percentage of the LH component mixed in the wavefunction.

From the k.p Hamiltonian, we can see that CB interacts with both HH and LH, but HH and LH do not interact with each other. Thus, one can easily imagine that if the wavefunction has more of the CB component (than the HH component), then through direct interaction, it would mix more with the LH component, leading to a larger LL splitting. With this picture in mind, we can now understand the relatively low g -factors in the wide-period limit compared to that in the bulk case. In the bulk case, the g -factor obtained is for the CB, whose wavefunction is dominated by the electron component. In sharp contrast, the wavefunction in the wide-period limit is dominated by the HH component. Since HH does not interact with LH directly, the wavefunction would not have much LH component, which explains the much smaller LL splittings (i.e., lower g -factors). Using a similar argument, we can also explain why the g -factors in the wide-period limit are much lower than that in the short-period limit, where our system is in the normal band alignment, and the wavefunction is dominated by the CB component.

We note that the above g -factor engineering principle (i.e., tuning g -factors by manipulating the electron component in the band) is considered in Step 1 of our two-step process. It is realized by changing the band gap between CB and HH in the SL samples. It also explains why the tuning range of g -factors is so wide for SL samples.

[Our changes]:

“”

3. The validation of the KP model? It is strange to me that the authors use a single set of fitting parameters for all the samples. The normal (1445) and inverted (1446) bandgap are different as the authors pointed out. Using the same band parameter for all three samples is still strange to me. Indeed this simplifies the story to only the variation of InAsSb thickness as discussed in SI note 2. Since the entire story is built on the KP model, it may be worth exploring more parameter space to make sure it is valid.

[Our reply]: The k.p model has proved to be very successful in understanding the physics and experiments in narrow-gap semiconductors and topological materials. In seminal work [Science **314**, 1757 (2006)], it predicted the presence of a quantum spin Hall insulator phase in HgTe quantum wells. In the following-up experiment [Science **318**, 766 (2007)], it precisely described the evolution of the band structures in HgTe/CdTe systems using a *unified* set of band parameters and taking the quantum well thickness as the only varying parameter. This is exactly what we have done in this work.

This approach has widely been used by others in the community. For example, Phys. Rev. B 80, 035303 (2009), Phys. Rev. B **94**, 245402 (2016), Phys. Rev. Lett. **119**, 056803 (2017), and Phys. Rev. B **101**, 075433 (2020), to list a few. In our previous work [Phys. Rev. B **95**, 045116 (2017)], we have also successfully explained the magneto-infrared (magneto-IR) experiment in InAs/GaSb systems using a single set of band parameters.

In addition, Reviewer #1 raises the concern about the choice of band parameters in both Comments (3) and (4). We will explain in detail how we determine our choice in reply to the next comment. Here, we would like to point out that our choice is either consistent with those provided by Vurgaftman *et al.* ([S6] in SI) or constrained by the experiment, and there is not much room to play around with our set of band parameters.

4. Furthermore, in the SI table, I list the band parameter. It is stated in the SI note 2 “we start with the parameters using the interpolation scheme between InSb and InAs but adjust the parameters in Tab. 1 to best fit the experiment.” However, there are many parameters which can be adjustable. As mentioned, the entire story is based on the KP model, if there are multiple solutions to this fits, it may change the story. Especially, the delta (spin-orbit coupling) energy for InAsSb is expected to be higher than InSb and InAs (Mayer W. et al. (ACS Appl. Electron. Mater. 2020, 2, 2351–2356), Ref. 18 and Ref. 33). It is contradicting table I. Moreover, ref. 6 in the SI has provided a table for InAsSb band parameters which is very different from the values of the fitting. Could the authors elaborate more on this point?

[Our reply]: We also thank Reviewer #1 for raising this point. We feel that we did not explain enough about the choice of the fitting parameters. Below, we will explain in detail and make the corresponding changes in the revised SI.

First, the band parameters for InSb are fully consistent with those provided by Vurgaftman *et al.* ([S6] in SI). For InAs_{1-x}Sb_x with $x = 52\%$, we consider the bowing effect in the alloy through the following relation

$$P_{InAs_{1-x}Sb_x} = P_{InAs}(1 - x) + P_{InSb}x - C_p x(1 - x),$$

where P denotes a band parameter, and C_p is the corresponding bowing parameter. As one can see here, a positive C_p value will inevitably make the parameter smaller than its binary constituents. Vurgaftman *et al.* suggested $C_\Delta = 1.2$ eV for the spin-orbit coupling energy Δ , which is close to $C_\Delta = 1.184$ eV we use. Therefore, we respectfully disagree with Reviewer #1’s comment that “the delta (spin-orbit coupling) energy for InAsSb is expected to be higher than InSb and InAs”.

Second, the band gap E_g of InAs_{0.48}Sb_{0.52} is calculated using a different bowing parameter from Vurgaftman *et al.* ([S6] in SI). It is measured to be $C_{E_g} = 0.83$ in our samples ([24] of the main text). In order to achieve a better fit to our data, we allow the Kane energy E_p and the valence band offset E_v to vary. The bowing effects for these two parameters are not discussed in [S6] of SI but reported in [Ref-1] for E_p and [Ref-

2] for E_v . By adjusting these two parameters, we can reach a global fit in good agreement with all three samples shown in the manuscript.

We summarize the above discussion in the table below. As one can see, the band parameters are highly constrained either by experiments or by literature, leaving out not much space to vary. We highlight the two fitting parameters E_p and E_v in red.

Table T1: Band parameters comparison between literature and our work. $\gamma_{1,2,3}$ are the modified Luttinger parameters, and they are related to the Vurgaftman's $\gamma'_{1,2,3}$

Luttinger parameters via $\gamma_1 = \gamma'_1 - \frac{E_p}{3E_g}$, $\gamma_2 = \gamma'_2 - \frac{E_p}{6E_g}$, $\gamma_3 = \gamma'_3 - \frac{E_p}{6E_g}$.

	InAs [S6]	InSb [S6]	InAs _{0.48} Sb _{0.52}	Bowing [S6]	This work
x	0.48	0.52			
a (Å, 5K)	6.05022	6.46913	6.26805	0	6.268
E_g (eV)	0.417	0.235	0.115	0.83 [24] 0.67	0.115
Δ (eV)	0.39	0.81	0.309	1.2	0.313
E_p (eV)	21.5	23.3	22.4	0	18.0
γ_1	2.81375	1.75036	2.26	0	2.26
γ_2	-0.09313	-1.02482	-0.578	0	-0.578
γ_3	0.60688	-0.02482	0.278	0	0.278
κ	-1.05979	-1.95816	-1.53	0	-1.53
E_v (eV)	-0.59	0	-0.28	0	-0.22

[Ref-2] P. T. Webster *et al.* J. Appl. Phys. **118**, 245706 (2015).

[Our changes]:

“”

5. The effective g-factor is obtained from fitting to a model instead of experimental data. It was surprising to me in the first place. But, I later realized that fitting the four-band model to the magneto-IR spectra will lead to very large errors due to not clearly defined transition between LLs. Is this due to the temperature effect? Did the authors also measure the magneto-IR spectra (figure 3) at 5K? What about the error bars to this fitting? Would it be possible to estimate the errors from the experiment data set when fitting with 8-band KP model?

Figure R1: Normalized transmission spectra, $T(B)/T(B = 0T)$, of sample 1445 at selected magnetic fields. Black dash lines are multi-Lorentzian fits to the data, corresponding to LL transitions. Low-lying transitions (or modes) are numbered in the same way as in Fig. 3 of the main text. Gray areas indicate the distorted spectral lineshape due to the presence of phonons.

[Our reply]: First, we would like to clarify that all experiments are performed at 5 K and most LL transitions (particularly interband LL transitions) in the magneto-IR spectra can be defined accurately. The seemingly large broadening in the false color map is due to the color saturations. In Fig. R1, we plot a few raw spectra after normalizing to the zero field. The LL transitions can be identified as absorption dips in the spectra with a Lorentzian lineshape. The black dash lines are multi-Lorentzian fits to the data at low energies. The fits capture the spectral lineshape of interband LL transitions well (modes 3,4,5...), with a typical error as small as ± 0.2 meV for each mode. Mode 2 exhibits an unusual broadening. But nevertheless, one can determine its central energy with great accuracy using Lorentzian fitting. The fit to mode 1, which is a cyclotron resonance mode, is affected by the presence of two strong modes at low energies (gray areas) whose energy position is independent of magnetic field. Overall, as one can see from Fig. 3 of the main text, the k.p fitting passes the center of most interband LL transitions (modes 2,3,4...), while the fit to mode 1 is the least accurate.

Second, Reviewer #1 precisely described our method in this work. That is, fitting the experimental data with the k.p model for the entire magnetic field range and then using the four-band model to fit the k.p results at $B \leq 1$ T to extract the g -factors. We choose this method because we are interested in g -factors in a practically accessible magnetic field where Zeeman splitting is usually small (even with a large g -factor) and LL transitions are not fully resolved. Therefore, it would be difficult to directly fit the experimental data with the four-band model.

Third, it is also difficult to extract the error between the k.p fitting and the experimental data. Instead, we can examine the robustness of the fitting by manually changing the

Figure R2: Calculated LL transition energies for sample 1445 using the k.p model with different E_p values. The calculation results are overlaid on the false color map of the magneto-absorption spectra shown in Fig. 3(a) of the main text.

main fitting parameter in the k.p calculation, i.e., the Kane energy E_p , and then compare the calculated LL transitions with the data. Figure R2 shows an example of the comparison when we either increase or decrease E_p from 18 eV to $E_p = 19$ eV (dash line) or $E_p = 17$ eV (solid line) while keeping all the other parameters unchanged in sample 1445. We find that both the dash and solid lines are not a good fit to the center of interband LL transitions but reasonably capture the boundaries (or broadening) of the transitions. Therefore, in the revised manuscript, we calculate the corresponding LLs using $E_p = 19$ eV and $E_p = 17$ eV and extract the associate g -factors. We use these g -factors to define the error bars for the g -factors in Fig. R3 (method 1), which are obtained using $E_p = 18$ eV.

As a result of the above three-step process, the extracted g -factors are as large as $g \approx 115$ in the short-period limit, slightly larger than those in the original manuscript.

[Our changes]: In the revised SI, we have added a new section about the fitting method and error estimation, including Figs. R1 and R2 above. We have also included Fig. R3 in the new Fig. 6 of the revised main text.

6. The result suggests that M1 is the dominant factor for the large effective g -factor. This is from the overlapping of CB and HH which makes sense to me around the small gap and HH is close to CB. Based on the assumptions, the model should be limited to a small gap with a small magnetic field which suggests around the normal-inverted band transition. However, the large effective g -factor is also observed for a normal band with a very large bandgap up to 120 meV. The large g value remains the same for the

Figure R3: Evolution of the effective g -factors for the CB, extracted using two different methods, as a function of the SL period.

Eg from 120 to 30 meV. This is beyond the assumptions of the four-band model.

[Our reply]: We respectfully disagree with Reviewer #1 that the four-band model is only valid when the CB and HH are close to each other. The assumption for the four-band model is that all other bands are far away from CB and HH so that we can treat their interactions as a perturbation. This is effectively the Lowdin perturbation approach [Ref-3], valid for both the degenerated and non-degenerated cases. Therefore, the four-band model is still applicable even if the band gap between the CB and HH is large. For example, it has been experimentally demonstrated that the four-band model is valid in Bi_2Se_3 , an archetypal topological insulator, with a band gap of ~ 300 meV [Phys. Rev. Lett. **114**, 186401 (2015); Phys. Rev. B **82**, 045122 (2010)].

Next, we explain why the g -factors are saturated in the short-period limit. As stated in reply to Comment (2), the SL is in the normal regime in this limit, and the wavefunction is dominated by the CB component. Therefore, the interaction between CB and LH leads to a large LL splitting. The saturation of the g -factors is due to the complete domination of the CB component in the wavefunction and the nearly constant energy gap between CB and LH.

Lastly, we should note that the saturation of the g -factors depends on its definition. The g -factors defined above with the four-band model (method 1) are proportional to the M_1 parameter, as shown in Eq. (3) of the main text. Such a g -factor is related to the splitting of LLs with the same index. However, as the g -factor increases, the splitting between LLs of the adjacent index can become dominant, giving rise to an alternative method (method 2) to define g -factors. That is, we follow a specific LL and define the g -factor from the splitting with the closest LL of the opposite spin regardless of its LL index. The extracted g -factors using these two methods are summarized in Fig. R3 as a function of the SL period. As one can see, the two methods agree well before entering the short-period limit. The difference in the short-period limit is caused by choosing a

different LL splitting to define the g -factor. We understand that this discussion may not be related to Reviewer #1's comment. But, it is helpful to complete our understanding of the g -factor behavior. Both methods support the realization of giant g -factors in our InAsSb/InSb SLs.

[Ref-3] R. Winkler, Spin-orbit Coupling Effects in Two-Dimensional Electron and Hole Systems, Springer-Verlag, Berlin (2003).

[Our changes]: In the revised SI, we have included several .

7. Lastly, it is also possible to directly measure the effective g -factor from the electrical transport properties. For example, Qu, F., et al. Nano Letters, 16(12), 7509–7513, Zijin Lei, et al., Phys. Rev. Research 3 023042 (2021) and ref. 33 etc. can measure the effective g -factor from the energy difference of the spin split LLs. It will provide a piece of more direct evidence to the claimed large g -factor. On the other hand, several potential applications of such superlattice structure as the authors mentioned in the main text will require electrical transport. Therefore, an electrical transport result may prove the usefulness of the InSb/InAsSb superlattice.

[Our reply]: We agree with Reviewer #1 that transport experiments would also be helpful and interesting. However, transport measurements are most effective in 2D samples such as quantum wells and heterostructures. The popular transport methods for determining g -factors include tilted magnet field method and point contact method (cited by Reviewer #1), both of which are restricted to 2D systems. One strategy is to produce short-period InAsSb/InSb quantum well samples and measure the g -factor. But, we find that the electron-hole wavefunction overlap is also suppressed by the presence of barriers above and below the quantum wells. It is a work in progress, but beyond the scope of this manuscript.

Other suggestions:

1. In Fig. 3b, sample 1446 shows an oscillating behavior in the energy direction between 60-240 meV. This is not seen in other samples. Any reasons for that?

[Our reply]: We thank Reviewer #1 for carefully reading our manuscript and pointing out the following three issues.

The oscillating behavior in Fig. 3(c) of the main text is a pure artifact due to the coarse magnetic field grid. In Fig. 3(c), the magnetic field is taken at 1 T step while it is 0.12 T in Fig. 3(a). We note that taking magneto-absorption spectra at 0.12 T step up to 17 T is very time and liquid helium consuming. We cannot afford it for every single measurement.

[Our changes]: We have commented on this issue in the revised caption to Fig. 3.

2. SI table I needs units.

[Our changes]: We have added units to the tables in the revised SI.

3. The labels in figure 4 are not very clear. I would replace green labels with another color.

[Our changes]: For better presentation, we ...

Reviewer #2 (Remarks to the Author):

This paper by Y. Jiang et al. reports the realization of giant Lande g-factor (104-110) in InAsSb/InSb SLs, which are designed/explained based on the virtual substrate technique, band gap tuning, and electron-hole wavefunction overlaps. The large g-factor is generally of great interest in a variety of applications including quantum devices as described in the abstract. The sample system and experimental/theoretical schemes seem to have solid basis, which are extensions of previous works done by some of the authors. I found, however, several problems in the manuscript; one of which would be critical and make the judge of significance difficult. Some of the points are listed below. Therefore, I do not recommend this paper to be published in Nature Communications in the present form.

We thank Reviewer #2 for carefully reviewing our manuscript and finding our work “generally of great interest.”

1. It is critical to add the error bars in the evaluation of g-factor, especially when the authors are claiming the record-large value. The error can appear from experimental broadening of magneto-IR absorption (apparent in presented figures), fitting procedure, and also through theoretical models. Please define the error and discuss the possible range of g-factor value in the manuscript.

[Our reply]: We thank Reviewer #2 for the constructive comment. We have followed the advice and determined the g-factor error bars through the following steps. We note

Figure R1: Normalized transmission spectra, $T(B)/T(B = 0T)$, of sample 1445 at selected magnetic fields. Black dash lines are multi-Lorentzian fits to the data, corresponding to LL transitions. Low-lying transitions (or modes) are numbered in the same way as in Fig. 3 of the main text. Gray areas indicate the distorted spectral lineshape due to the presence of phonons.

that since a related question has also been raised by Reviewer #1 in Comment (5), we duplicate Figs. R1-R3 and some statements here from our reply above.

(1) First, we extract the Landau level (LL) transition energies from the magneto-infrared (magneto-IR) absorption spectra. Figure R1 shows a few raw spectra after normalizing to the zero field. The LL transitions can be identified as absorption dips in the spectra with a Lorentzian lineshape. The black dash lines are multi-Lorentzian fits to the data at low energies. The fits capture the spectral lineshape of interband LL transitions well (modes 3,4,5...), with a typical error as small as ± 0.2 meV for each mode. Mode 2 exhibits an unusual broadening. But nevertheless, one can determine its central energy with great accuracy using Lorentzian fitting. The fit to mode 1, which is a cyclotron resonance mode, is affected by the presence of two strong modes at low energies (gray areas) whose energy position is independent of magnetic field. However, once we determine all the interband LL transitions, the energy of this CR mode is known and can be deduced from the LL fan diagram shown in Fig. 3(b,d) of the main text.

(2) Second, we recognize that in our k.p fitting, the main fitting parameter is the Kane energy E_p . When using $E_p = 18$ eV (Fig. 3(a) of the main text), the calculated LL transitions pass the centers of most interband absorption modes and exhibit a good fit. In Fig. R2, we examine the robustness of the fitting by manually increasing or decreasing E_p to $E_p = 19$ eV (dash line) or $E_p = 17$ eV (solid line) while keeping all the other parameters unchanged in sample 1445. We find that both the dash and solid lines are not a good fit to the centers of interband LL transitions but reasonably capture

Figure R2: Calculated LL transition energies for sample 1445 using the k.p model with different E_p values. The calculation results are overlaid on the false color map of the magneto-absorption spectra shown in Fig. 3(a) of the main text.

Figure R3: Evolution of the effective g -factors for the CB, extracted using two different methods, as a function of the SL period.

the boundaries (or broadening) of the transitions. Therefore, we conclude that the error bar of E_p for the k.p fitting to the sample 1445 data is ± 1 eV.

(3) Third, given the above error bar, we calculate the corresponding LLs using $E_p = 17$ eV and $E_p = 19$ eV and extract the g -factors in the low field regime ($0 \leq B \leq 1$ T). This way, we can define the upper and lower bound of the g -factors. The results are shown in Fig. R3, while the details will be explained in reply to Comment (4).

As a result of the above three-step process, the extracted g -factors are as large as $g \approx 115$ in the short-period limit, slightly larger than those in the original manuscript.

[Our changes]: In the revised Supplementary Information (SI), we have added a new section about the fitting method and error estimation, including Figs. R1 and R2 above. We have also included Fig. R3 in the new Fig. 6 of the revised main text.

2. Please revise the right panel of Fig. 1, and depict quantitatively how much wavefunction overlap is expected at which SL period.

[Our reply]: We thank Reviewer #2 for the constructive comment. Figure 1 of the main text is meant to schematically demonstrate our overall strategy for g -factor engineering, whereas quantitative calculations of wavefunction overlap require substantial theoretical background information. Therefore, based on the reviewer's advice, we have included a new figure (Fig. R4) as a part of Fig. 6 in the revised manuscript to demonstrate the electron-hole (e-h) wavefunction overlap as a function of the SL period using the band parameters extracted from the experiment. We note that due to the quantization effect, there are many subbands in our SL systems, and we only consider the wavefunction overlap of interest, that is, the overlap between the conduction (CB) and heavy-hole (HH) subbands.

Figure R4: Calculated in-plane e-h wavefunction overlap between CB and HH as a function of the SL period. The calculation is performed at 1 T using the band parameters extracted from the experiment. Insets: Band evolution from the normal state to the inverted state as increasing the SL period. Here, LH stands for the light-hole band.

[Our changes]: We have included Fig. R4 as a part of new Fig. 6 in the revised manuscript to quantitatively depict the e-h wavefunction overlap as a function of the SL period.

3. I believe that “high-resolution cross-sectional TEM” nowadays means that with atomic resolution. Please replace Fig. 2 with the one showing the atomically sharp interface. If some disorder is unavoidable in the growth, add some discussion on the possible effects of disorder in the g-factor engineering performed in this work.

[Our reply]: We have performed extensive growth and structural characterizations of $\text{InAs}_{1-x}\text{Sb}_x/\text{InSb}$ SLs in our previous works [21, Ref-4]. In Ref. [21] of the main text, a high-resolution cross-sectional transmission electron microscopy (TEM) image is shown for $\text{InAs}_{0.48}\text{Sb}_{0.52}/\text{InSb}$ metamorphic SLs grown by the same method but with a slightly different period. Here, we reproduce Fig. 4(a) of Ref. [21] in Fig. R5 for Reviewer #2’s reference.

As one can see from Fig. R5, our SL samples exhibit a dislocation-free periodic structure but with some interface roughness and compositional disorder. The effects of interface roughness have been discussed in Ref. [21] and modeled as a local random change in the SL layer thickness. Such a lateral thickness fluctuation does not directly contribute to the LL transition energies but leads to the broadening of the transitions. The broadening is somewhat suppressed in the ternary-binary SLs ($\text{InAsSb}/\text{InSb}$) compared with the ternary-ternary ones. In addition, the composition of $\text{InAs}_{0.48}\text{Sb}_{0.52}$ layer in our SLs is chosen to be close to the minimum of the band gap bowing curve. It

Figure R5: High-resolution cross-sectional TEM image of a similar InAsSb/InSb SL sample. Brighter areas correspond to the InSb layers. Adopted for Ref. [21] of the main text.

makes the material parameters less sensitive to the fluctuations of the composition.

[Ref-4]: G. Belenky *et al.* *Appl. Phys. Lett.* **117**, 250501 (2020).

[Our changes]: In the revised manuscript, we replace “high-resolution cross-sectional TEM image” with “cross-sectional TEM image” to avoid confusion. We also include a new paragraph in Methods – Sample growth to describe the possible effects of interface roughness.

4. Please explain Fig. 5(d) (actually not labeled) in more detail. Why the g -factor does not change at all for the InAsSb thickness of 2 to 3 nm (also for 5.5 to 6 nm) even when the bandgap keeps changing?

[Our reply]: We thank Reviewer #2 for raising this question. In the original manuscript, when we change the SL period (i.e., the band gap), we first fix the g -factor and evaluate by eye if the four-band model can fit the k.p results in the low field regime ($B \leq 1$ T). We find that the fits are reasonable in the short-period and wide-period limits by changing the band gap alone. Therefore, the g -factor in Fig. 5(d) of the original manuscript does not change for the InAsSb thickness of 2 to 3 nm and 5.5 to 6 nm.

However, this approach introduces errors in the fitting process. In the revised manuscript, we follow the advice of Reviewer #2 Comment (1) and perform a rigorous fitting and error analysis as described above. The extracted g -factors as a function of the SL period are shown in Fig. R3 as method 1. Here, in the short-period limit (i.e., for thinner InAsSb), the g -factors exhibit a saturation behavior as decreasing the SL period. In the wide-period limit (i.e., for thicker InAsSb), the g -factors continue to decrease as increasing the SL period, but the trend gradually slows down.

We can understand the g -factor behavior of Fig. R3 (method 1) from the corresponding SL band structure, which is shown schematically in the inset to Fig. R4 as a function of the SL period. As we know, to introduce splittings between LLs, one needs to include the interactions between the CB/HH band and the LH/split-off(SO) band. For simplicity, here we consider the LH band. We can qualitatively relate the LL splittings with the percentage of the LH component mixed in the wavefunction.

From the k.p Hamiltonian, we can see that CB interacts with both HH and LH, but HH and LH do not interact with each other. Therefore, one can easily imagine that if the wavefunction has more of the CB component (than the HH component), then through

direct interaction, it would mix more with the LH component, leading to a larger LL splitting. In the wide-period limit, the band is inverted, and the wavefunction of interest is dominated by the HH component. Since HH does not interact with LH directly, the wavefunction would not have much LH component, which explains the much smaller LL splittings (i.e., lower g -factors). As the SL period increases, the wavefunction becomes more and more HH dominant. Thus, the g -factor decreases monotonically.

Using a similar argument, we can also explain why the g -factors in the short-period limit are much larger. It is because the SL is now in the normal band alignment, and the wavefunction is dominated by the CB component. Therefore, the interaction between CB and LH leads to a large LL splitting. The saturation of the g -factors in the short-period limit is due to the complete domination of the CB component in the wavefunction and the nearly constant energy gap between CB and LH.

[Our changes]: We have modified Fig. 5(c) of the main text to illustrate the energies of CB, HH, and LL to demonstrate the dominant CB component in the wavefunction in the normal regime and the domination of HH in the inverted regime. The revised figure is now a part of new Fig. 6. We have also included a description of the extracted g -factors as a function of the SL period (as described above) in the Discussion section of the revised manuscript.

Lastly, we further study the g -factor behavior in the short-period limit. We note that the g -factors defined above with the four-band model (method 1) are proportional to the M_1 parameter, as shown in Eq. (3) of the main text. Such a g -factor is related to the splitting of LLs with the same index. However, as the g -factor increases, the splitting between LLs of the adjacent index can become dominant, giving rise to an alternative method (method 2) to define g -factors. That is, we follow a specific LL and define the g -factor from the splitting with the closest LL of the opposite spin regardless of its LL index. The extracted g -factors using these two methods are summarized in Fig. R3 as a function of the SL period. As one can see, the two methods agree well before entering the short-period limit. The difference in the short-period limit is caused by choosing a different LL splitting to define the g -factor. We understand that this discussion may not be related to Reviewer #2's comment. But, it is helpful to complete our understanding of the g -factor behavior. Both methods support the realization of giant g -factors in our InAsSb/InSb SLs.

[Our changes]: We replace Fig. 5(d) with Fig. R3 in the revised manuscript and add the discussion of g -factor definition method 2 in the Discussion section.

Other changes to the manuscript:

We corrected several typos in the manuscript. The essential ones are in Table 2 of SI: For $\text{InAs}_{0.48}\text{Sb}_{0.52}$, $C_{11} = 758.5$ GPa and $C_{12} = 412.5$ GPa; for InSb, $a_v = -0.36$ eV (missing “-” sign).

Reviewers' Comments:

Reviewer #1:

Remarks to the Author:

Thanks to the authors' reply. The error estimation of the g-factor provides essential information in the supplementary info. At least for the g-factor, readers can have a rough idea what the range of errors. However, a few questions remain for this version of the manuscript.

1. One of the key points of the manuscript is the giant g-factor in the SL structure. Figure 5 (b) shows a range of g-factor from 20 to 110 by tuning the period of the SL structure but is purely based on theoretical calculation. Therefore, the connection between theoretical to experimental results is crucial. Since the thickness ratio for 1445/1446 is very close to 1.8, one can try to point out that sample 1445/1446 will be at which point of figure 5(b). It may provide a good reference point for readers linking the experimental data in the final discussion.

2. Following the previous question, the samples 1445 and 1446 have different SL periods of 160 and 147 (perhaps, 1444 is also very close to 1.8 in thickness ratio and period of 213). Are they different enough to provide experimental evidence such that the g-factor is tunable by the SL period? Can these three samples provide a trend to g-factor values versus the SL periods?

3. Since the claimed giant g-factor 104 is from sample 1444 that is only presented in the supplementary info. Do samples 1445 and 1446 have a lower or higher g-factor? It will be very helpful to provide the experimental g-factor value for 1445/1446.

4. The discussion of figure 5 in the main text is based on the period variation but the variable is based on InSbAs thickness. It is a bit confusing since the period is based on how many repetitions of InSb/InSbAs. In the main text, the authors do not explain but for non-expert readers, it will be very helpful to have a few sentences to justify why InSbAs can represent period variation.

5. For spin-orbit strength of InAs, InSb, and InSbAs in the band fitting, it has been shown in several papers that InSbAs quantum wells/nanowires have a higher spin-orbit coupling strength. Therefore, I do not agree with the authors. It will be still hard for me to understand that the spin-orbit strength behaves much differently for the same materials in the SL structure and non-SL structures. Here are some experimental references to show that InSbAs have a higher SOC than InSb and InAs. Therefore, do the fitting parameters in table I(SI) remain valid?

Ref. for SOC in InSbAs:

(1) Moehle, C. M., et al. InSbAs Two-Dimensional Electron Gases as a Platform for Topological Superconductivity. *Nano Letters*, 21(23), 9990–9996.

(2) Sestoft, J. E., et al. Engineering hybrid epitaxial InAsSb/Al nanowires for stronger topological protection. *Physical Review Materials*, 2(4), 044202.

6. There is a typo in figure 3 caption, "The low-lying transitions are numbered in the same sequence as in Figure 3c,d. TLH denotes a weak transition from LH to HH LLs, and T₋ marks a magnetic field independent spectral feature that we believe is not related to the band structure under this study." Figure 3c,d should be figure 2c, d

Reviewer #2:

Remarks to the Author:

The authors have prepared reasonable replies to the comments, mostly, with the corresponding revisions in the main text and SI. Now I would say that this paper can be accepted for publication in *Nature Communications*.

REVIEWER COMMENTS

Reviewer #1 (Remarks to the Author):

Thanks to the authors' reply. The error estimation of the g -factor provides essential information in the supplementary info. At least for the g -factor, readers can have a rough idea what the range of errors. However, a few questions remain for this version of the manuscript.

We thank Reviewer #1 for recognizing the improvements made in the previous revision and providing additional constructive suggestions/comments. Below we answer his/her questions point by point.

1. One of the key points of the manuscript is the giant g -factor in the SL structure. Figure 5 (b) shows a range of g -factor from 20 to 110 by tuning the period of the SL structure but is purely based on theoretical calculation. Therefore, the connection between theoretical to experimental results is crucial. Since the thickness ratio for 1445/1446 is very close to 1.8, one can try to point out that sample 1445/1446 will be at which point of figure 5(b). It may provide a good reference point for readers linking the experimental data in the final discussion.

[Our reply]: First, we want to emphasize that the g -factors shown in Figure 5b are NOT purely based on theoretical calculation. It is based on realistic band parameters determined by fitting the well-established eight-band k .p model to the accurate magneto-infrared spectroscopy data. We have achieved excellent agreements for the three different samples measured with the corresponding model calculation using a single set of band parameters. Therefore, these parameters are experimentally verified and can be effectively used to describe the measured samples and predict the properties of other superlattices (SLs) with the same composition but different period (*i.e.*, with a different InAsSb thickness since the InAsSb/InSb thickness ratio within one period is a constant 1.8). Such an approach has been widely adopted for interpreting the electronic structures of III-V semiconductors and topological materials, as discussed in the previous Reply.

Second, we agree with Reviewer #1 that labeling out the three SL samples (1444, 1445, and 1446) in Figure 5b can provide a good reference point for readers. The InAsSb thickness in each period of the three samples is 3 nm, 4 nm, and 5 nm, and the extracted g -factor is 104, 78, and 55, respectively.

[Our changes]: In the revised manuscript,

(1) We label out the three SL samples in Figure 5b. Since the discussion in the main text is mostly about the four-band model, which defines the g -factor using Eq. 2 (method 1), we extract the g -factor in the three samples using method 1. We also modify the caption to Figure 5b accordingly.

(2) We add the following sentence in the last paragraph on Page 9

“The extracted band parameters can well describe the electronic structure of all three samples measured and predict the properties of similar SLs with different periods. The resulting band edge, ...”

2. Following the previous question, the samples 1445 and 1446 have different SL periods of 160 and 147 (perhaps, 1444 is also very close to 1.8 in thickness ratio and period of 213). Are they different enough to provide experimental evidence such that the g -factor is tunable by the SL period? Can these three samples provide a trend to g -factor values versus the SL periods?

[Our reply]: Based on Reviewer #1's comments, we feel that we should improve the description of our sample structures and the strategy to control the g -factors in these samples. Table R1 summarizes the structure information of the three SL samples studied in this work.

Table R1: Sample structure information. x is the alloy composition of Sb in InAsSb, N is the total number of periods in SLs, and L_t is the total thickness of SLs.

Sample number	x	t_{InAsSb} (nm)	t_{InSb} (nm)	$t_{\text{InAsSb}}/t_{\text{InSb}}$	N	L_t (μm)
1444	0.52	3	1.69	1.8	213	1.0
1445	0.52	4	2.25	1.8	160	1.0
1446	0.52	5	2.82	1.8	147	1.15

From the Table, we can see that the number of periods for samples 1444, 1445, and 1446 is 213, 160, and 147, respectively. These numbers need to be large enough so that the samples can be regarded as SLs rather than quantum wells. Consequently, when the number of periods is sufficiently large ($>$ several tens), the electronic structure will stop changing with the number of periods or the total thickness. Therefore, in our SLs, the number of periods does not tune the g -factors and the electronic structures.

Instead, the structure of each period (including an InAsSb layer and an InSb layer) is the major tuning parameter for the electronic structures and g -factors in our SL samples. The underlying mechanism is that the conduction and valence band alignment between layers can be manipulated through the quantum confinement effect, which is realized via varying the layer thicknesses during the growth. Of course, there are different ways to manipulate the electronic structure in metamorphic InAsSb/InSb SLs. For example, one can fix the InSb layer but change the thickness of the InAsSb layer in each SL period or the other way around. In this work, we choose to manipulate the electronic structure and g -factor by changing the thickness of both layers but keeping their thickness ratio $t_{\text{InAsSb}}/t_{\text{InSb}}$ at 1.8. A constant thickness ratio is required to keep the SL strain compensated and stabilize the structure. Hence, we use the thickness of the InAsSb layer (which are integer numbers in the three samples studied) to represent the thickness of each SL period in Figure 5.

Again, by fitting the experimental results with our model calculations, we extract a g -factor of 104, 78, and 55 for samples 1444, 1445, and 1446, respectively. One can find the trend in the g -factor values versus the SL periods for the three samples measured in the revised Figure 5b.

[Our changes]: In the revised manuscript,

(1) We label out the three SL samples in Figure 5b and illustrate the trend in the g -factor values versus the SL periods.

(2) We add Table R1 to the Methods section in the main text to better describe the three SL samples studied. Additional descriptions are also added with the Table.

(3) We add the following sentence in the Methods section (Page 12)

“We note that the number of periods N is sufficiently large in our SL samples such that the electronic structure and g -factor are independent of N but can be tuned by changing the thickness of each SL period.”

3. Since the claimed giant g -factor 104 is from sample 1444 that is only presented in the supplementary info. Do samples 1445 and 1446 have a lower or higher g -factor? It will be very helpful to provide the experimental g -factor value for 1445/1446.

[Our reply]: This comment is in line with the above two comments. Following Reviewer #1's suggestion, we provide the experimental g -factors for samples 1445 and 1446 in the main text.

[Our changes]: We add the following sentence in the revised manuscript (Page 9-10)

“Specifically, the extracted g -factor for samples 1445 and 1446 is 78 and 55, respectively.”

4. The discussion of figure 5 in the main text is based on the period variation but the variable is based on InSbAs thickness. It is a bit confusing since the period is based on how many repetitions of InSb/InSbAs. In the main text, the authors do not explain but for non-expert readers, it will be very helpful to have a few sentences to justify why t_{InSbAs} can represent period variation.

[Our reply]: We thank Reviewer #1 for carefully reading our figures and pointing out this issue. As mentioned in reply to Comment (2), when tuning the electronic structure and g -factor in our SLs, we choose to change the thickness of both InAsSb and InSb layers while keeping their thickness ratio $t_{\text{InAsSb}}/t_{\text{InSb}}$ at 1.8 (see Table R1). In this way, we keep the SL strain compensated. Therefore, changing t_{InAsSb} is equivalent to changing the SL period. We use t_{InAsSb} as the x -axis in Figure 5 simply because they are integers (3 nm, 4 nm, and 5 nm) in the three samples studied. For better presentation, we have followed the reviewer's advice to clarify this issue in the main text.

[Our changes]: In addition to the changes made for Comment (2), we include the following expression in the main text (Page 9) and the caption to Figure 5 to remove the confusion

“... the SL period ($t_{\text{period}} = t_{\text{InAsSb}} + t_{\text{InSb}} = 1.6t_{\text{InAsSb}}$).”

We also define the meaning of ‘the SL period’ in the revised Introduction (Page 4)

“... the period of InAsSb/InSb SLs (that is, the thickness of the InAsSb and InSb layers in one period), ...”

5. For spin-orbit strength of InAs, InSb, and InSbAs in the band fitting, it has been shown in several papers that InSbAs quantum wells/nanowires have a higher spin-orbit coupling strength. Therefore, I do not agree with the authors. It will be still hard for me to understand that the spin-orbit strength behaves much differently for the same materials in the SL structure and non-SL structures. Here are some experimental references to show that InSbAs have a higher SOC than InSb and InAs. Therefore, do the fitting parameters in table I(SI) remain valid?

Ref. for SOC in InSbAs:

(1) Moehle, C. M., et al. InSbAs Two-Dimensional Electron Gases as a Platform for Topological Superconductivity. *Nano Letters*, 21(23), 9990–9996.

(2) Sestoft, J. E., et al. Engineering hybrid epitaxial InAsSb/Al nanowires for stronger topological protection. *Physical Review Materials*, 2(4), 044202.

[Our reply]: We thank Reviewer #1 for bringing this issue to our attention. We find that this seems to be an unresolved puzzle. In addition to the two references Reviewer #1 provides, a negative bowing parameter for spin-orbit coupling (*i.e.*, higher SOC) has been suggested in an earlier experiment [Cripps, S. A., *et al.* *APL* **90**, 172106 (2007)]. On the other hand, there are other experiments supporting a positive bowing parameter, such as [24] of the main text, [7] of the Supplementary Information, and [O. Berolo and J. C. Woolley, *Proceeding of the 11th International Conference on the Physics of Semiconductors (ICPS)*, Warsaw, 1972, p.1420]. It is not within the scope of this work to settle this issue. Nevertheless, the reason for us to choose a positive bowing parameter is to be self-consistent with [24] of the main text, where the experiments were done on our samples, and the results support a positive bowing parameter.

Moreover, we note that the difference between the SOC bowing parameters only produces a negligible effect on the band structure of our SL samples. It is because the conduction (CB) and valence bands of interest are mostly composed of light-hole (LH), heavy-hole (HH), and CB components, and the split-off band is far away and only affects the band structure through perturbation. In Figure R1, we compare the band structure for the $t_{\text{InAsSb}} = 4$ nm case using SOC energy of 600 meV (suggested in [18] of the main text) and 313 meV (this work). It is evident that the band structure, particularly CB and HH, has barely changed, and therefore the band parameters used in this work provide an effective description of the SLs studied.

Figure R1: Calculated band structure of the SL sample with $t_{\text{InAsSb}} = 4$ nm using different SOC energies. The band structure, particularly CB and HH, barely changes with (nearly) doubling the SOC energy.

[Our changes]: We add the following statements in the revised Supplementary Information to clarify this issue (Page 6-7)

“Lastly, it is worth noting that for the split-off band gap Δ in $\text{InAs}_{0.48}\text{Sb}_{0.52}$, we consider the bowing effect [6] with a positive bowing parameter. Though both a positive [7, 10, 11] and negative [12–14] bowing parameter have been reported in the literature, we choose a positive value to be self-consistent with that in Ref. [11] measured in similar SL samples. We also find that because the split-off band is far away in energy and only affects the band structure through perturbation, the choice of bowing parameter for Δ has a negligible effect on the model fitting in the main text.”

We cite the relevant references accordingly, including the two suggested by Reviewer #1.

6. There is a typo in figure 3 caption, “The low-lying transitions are numbered in the same sequence as in Figure 3c,d. TLH denotes a weak transition from LH to HH LLs, and T_ marks a magnetic field independent spectral feature that we believe is not related to the band structure under this study.” Figure 3c,d should be figure 2c, d

[Our reply]: We thank Reviewer #1 for pointing out the typo. We have corrected it correspondingly.

Other changes to the manuscript:

- (1) Update Figure 1d,e. The previous plots have the correct band gap but calculated using a different energy offset from Figure 2. For consistency, we update it.
- (2) We correct a typo in the main text (Page 6, highlighted).
- (3) We have revised the format of the references based on the Nature Communications formatting instructions.

Reviewers' Comments:

Reviewer #1:

Remarks to the Author:

Thanks to the authors' reply. I am satisfied with the revised manuscripts and the manuscript should be considered to be published in Nature Communication.